# Artificial Intelligence in Food Safety: A Decade Review and Bibliometric Analysis

**DOI:** 10.3390/foods12061242

**Published:** 2023-03-14

**Authors:** Zhe Liu, Shuzhe Wang, Yudong Zhang, Yichen Feng, Jiajia Liu, Hengde Zhu

**Affiliations:** 1School of Management, Henan University of Technology, Zhengzhou 450001, China; 2School of Computing and Mathematical Sciences, University of Leicester, Leicester LE1 7RH, UK

**Keywords:** artificial intelligence, food safety, bibliometric review, CiteSpace

## Abstract

Artificial Intelligence (AI) technologies have been powerful solutions used to improve food yield, quality, and nutrition, increase safety and traceability while decreasing resource consumption, and eliminate food waste. Compared with several qualitative reviews on AI in food safety, we conducted an in-depth quantitative and systematic review based on the Core Collection database of WoS (Web of Science). To discover the historical trajectory and identify future trends, we analysed the literature concerning AI technologies in food safety from 2012 to 2022 by CiteSpace. In this review, we used bibliometric methods to describe the development of AI in food safety, including performance analysis, science mapping, and network analysis by CiteSpace. Among the 1855 selected articles, China and the United States contributed the most literature, and the Chinese Academy of Sciences released the largest number of relevant articles. Among all the journals in this field, *PLoS ONE* and *Computers and Electronics in Agriculture* ranked first and second in terms of annual publications and co-citation frequency. The present character, hot spots, and future research trends of AI technologies in food safety research were determined. Furthermore, based on our analyses, we provide researchers, practitioners, and policymakers with the big picture of research on AI in food safety across the whole process, from precision agriculture to precision nutrition, through 28 enlightening articles.

## 1. Introduction

Artificial intelligence (AI), as a far-reaching emerging technology, has experienced birth, ups, and downs, and the harvest, not only impacting our personal lives but also essentially transforming how firms make decisions [1]. Machine learning (ML) is currently a main branch of AI, integrating probability theory, statistics, and convex optimization to resolve the problems of computer vision, speech recognition, natural language processing, robot control, etc. [2]. Compared to other ML techniques, deep learning (DL) reveals excellent performance in image recognition, speech recognition, molecule prediction, particle accelerator data analysis, brain circuit reconstruction, etc. [3]. It is known that AI has three pillars, namely, data, algorithms, and computing power. By contrast, the food safety system has arisen as one of the most important application scenarios that use data-intensive approaches to drive the sustainable development of human beings and minimize its environmental impact.

Food safety is extremely important for human health and survival and deserves more advanced technologies to protect both consumers from foodborne illness and firms from reputational damage [4]. AI and big data, regarded as the fourth industrial revolution, already have a significant impact on the food industry by increasing food production, quality, and nutrition, and reducing resource consumption and waste [5]. Furthermore, recent studies have explored AI-based methods to deal with dietary problems that usually lead to chronic diseases such as hypertension [6].

Several reviews exist on AI applications in farm management, food processing, and food nutrition [7,8,9,10]. Still, panoramic scanning is scarce. On the other hand, bibliometric methods can explore the evolution and landscape of research, thus capturing the historical trajectory and future trends over time [11]. In this study, we make the following contributions:We selected 1855 articles as research samples to explore which AI technologies were applied for a sustainable food system from farm to fork;Our review elaborated on the development trend and current research hotspots on AI technologies in food safety and predicted the future research direction;This review should be helpful for researchers and practitioners to comprehensively understand the application status of AI technologies in the food sector;We have elaborated on the countries, institutions, and journals that have contributed much to the research on AI technologies in food safety.

The rest of this article is arranged as follows: First, we briefly outlined the development of AI in Section 2. Second, we outlined the overall research process in Section 3. Third, we portrayed the distribution in related subfields (e.g., food science technology, environmental science, remote sensing, nutrition dietetics information, etc.) and virtualized the science mapping (e.g., co-citation analyses and co-authorship analyses) in Section 4. Then, we conducted a network analysis (e.g., keyword co-occurrence analysis and cluster analysis) in Section 5. Finally, the advantages derived from the implementation of AI in food safety were listed, as well as the future expectations in the domain of the whole process.

## 2. Theoretical Background

AI has enabled computer systems or algorithms to learn insight, knowledge, and patterns from data and perform specific tasks without explicitly being programmed [12]. Machine learning is a subcategory of AI that implements intelligence. Deep learning, a subfield of machine learning, has revolutionised many domains of machine intelligence, such as computer vision, machine translation, and medical diagnosis. This technique enables machines to achieve competitive or even superior performance over humans. Due to the significant advantages in pattern recognition and image processing, CNN-based approaches have been the most popular architectures for food classification, quality detection, and nutrition evaluation [13].

Convolutional neural networks (CNNs) have especially become standard models for image recognition since AlexNet [14] significantly outperformed the existing contemporary approaches on the ImageNet LSVRC-2010 dataset and achieved absolute advantages in the ILSVRC-2012 competition. CNNs extract features and recognize patterns from the input images by sliding filters across the image. There has been a lot of interest in designing deeper and more complex CNN architectures to improve the representation capability further. State-of-the-art CNNs, such as DenseNet [15], ResNet [16], and VGGNet [17], have demonstrated that the increased depth has led to significant performance improvement. In particular, the residual skip connections introduced the problem of vanishing gradients and the degradation problem, allowing for an extremely deep architecture design.

Despite this, this approach inevitably increases computational costs and memory requirements. Much effort has been made to strike an excellent trade-off between model performance and computation efforts. CNNs built on depth-wise separable convolution and group convolution, such as ShuffleNet [18], ResNeXt [19], and MobileNet [20], have raised the possibility of mobile applications performed on a platform with limited computational resources. These networks are designed to pursue the best accuracy with fewer parameters and lower inference latency. More recently, neural architecture search (NAS) has demonstrated its potential in automatically designing CNNs that are comparable or even superior to those CNNs manually designed by human experts. Representative work includes EfficientNet [21], MnasNet [22], and NASNet [23], each of which utilises reinforcement learning to discover the optimal architecture from a search space of candidates. The focus of CNN architecture design has shifted from designing deeper and more complex networks to designing more efficient networks with high performance, either manually or automatically.

The above CNNs with powerful representation capability are further generalised to various domains using transfer learning techniques. By pre-training these networks on a large-scale dataset such as ImageNet with 1000 classes, these networks are equipped with prior knowledge and converge faster on the dataset of the target domain during fine-tuning. Transfer learning is constructive, especially when the dataset size is small. With a limited amount of data, the network might be unable to learn meaningful features. The pre-training process on a large-scale dataset allows CNNs to inherit well-trained low-dimensional filters that require many data to train.

Since food safety involves multiple disciplines, including environmental science, food science, economics, and agriculture [24], we strived to cover the following topics in this study: exploring the current AI-based applications and examples in the food industry, discovering the important research hotspots and pieces of literature, summarizing the trajectories and trends of AI development in food safety, identifying potential gaps, and creating a guideline for further research.

## 3. Method and Data

### 3.1. Method and Software

Our literature review has two steps: a systematic visualised review and a bibliometric analysis. In contrast to the prior studies [25,26] in AI research using a natural language processing toolkit or heuristic approach, we analysed the relevant data and the visual networks by CiteSpace to explore historical origins, evolutionary trajectories, and future trends in our research.

### 3.2. Sample

In the process of literature collection, we used SCI Expand search in the Core Collection database of WoS (Web of Science). Even though there are several bibliographic databases, such as Scopus, Google Scholar, Dimensions, Microsoft Academic, etc., WoS was more suitable for large-scale bibliometric analysis with high reliability [27]. Through the research on the previous papers in relevant fields, we determined the search formula as follows: TS = (“food safety” OR “foodborne illness” OR “nutrition” OR “food security” OR “food quality”) AND (“Artificial intelligence” OR” data clustering” OR “data mining” OR “neural network*” OR “Bayes* network” OR “semantic segmentation” OR “supervised learning” OR “unsupervised clustering” OR “feature* selection” OR “machine learning” OR “feature* extraction” OR “expert* system*” OR” deep learning” OR “big data”).

A total of 2378 documents were obtained according to the literature review search process. Based on bibliometric analysis [28,29], when conducting literature retrieval in WoS, we set three criteria for sample screening, namely “time span”, “document type”, and “WoS categories”. Then, we set the published years of the collected documents to the period from 2012 to 2022 and limited the types of documents to articles and reviews. After that, through the screening of WoS categories, records in categories less relevant to the focus of this study were excluded. Finally, 1855 primary literature samples of research on AI in the food safety field from 2012 to 2022 and 55,282 references to these pieces of literature were generated.

### 3.3. Analyses

We divided the selected 1855 samples into 15 fields, such as food science and technology, environmental science, remote sensing, nutrition dietetics, etc. After the descriptive analysis, we deployed science mapping [30], including co-citation and keyword occurrence analyses. Then, we further screened 143 highly cited and high-quality articles by network metrics and gained seven thematic clusters. Finally, 28 articles were selected and studied to present the AI applications in food safety across the whole process from precision agriculture to precision nutrition, as shown in Figure 1.

## 4. Analysis and Results

### 4.1. Current Status of Al Research

#### 4.1.1. Annual Trends

A number of publications have been released since 2019, as shown in Figure 2, indicating explosive interest in this subject. Due to the constant progress of AI technology, more and more applications are being employed to predict crop yields, control food quality, and reduce foodborne illness. The increasing trend seems set to continue, and more studies may emerge.

#### 4.1.2. Distribution of Publications

Figure 3 shows the analysis of the discipline category of the literature pieces. After analysing the discipline categories of all the selected pieces of literature, the top five discipline categories with the highest number of publications were food science technology (410 articles), environmental sciences (280 articles), remote sensing (215 articles), nutrition dietetics (193 articles), and imaging science information systems (179 articles). This indicated that the interdisciplinary integration characteristics of AI in food safety were evident. Furthermore, these results implied that the research on AI had expanded into various fields. However, there are still some significant gaps in the depth and breadth of these fields.

#### 4.1.3. Publication Timeline

In order to specifically describe the research on AI in food safety, we chose the five most popular subfields and then drew the publication timeline, as shown in Figure 4. The annual publications in these five fields presented a gradually increasing trend from 2012 to 2022, but the number of published papers before 2018 fluctuated to varying degrees. Since 2018, three fields, namely food science technology, environmental sciences, and nutrition dietetics, have been growing rapidly. This trend meant rapid expansion and recent advances in AI technologies. In addition, we noticed a declining trend in the fields of remote sensing, imaging science, and photographic technology after 2021. In summary, although AI in food safety increased attention in general, there were still significant differences in each subfield concerning AI development.

### 4.2. Co-Citation Analyses

#### 4.2.1. Author Co-Citation Analysis

Author co-citation analysis depicted the co-citation author network generated by CiteSpace to show the interrelationships among cited authors and identify the authors with strong influence [31]. Figure 5 demonstrates the co-citation network of AI in food safety research during 2012–2022. To show the author co-citation network clearly, we just displayed the authors with the most cited references in each time slice. There were 421 nodes and 1892 links in the author co-citation network. These nodes were plotted with citation tree rings containing several time slices. The citation tree rings’ thickness indicates the authors’ citation frequency in the corresponding time slice.

#### 4.2.2. Reference Co-Citation Analysis

Reference co-citation networks represented the structure and development of a field in detail, which was composed of the nodes and connections among co-cited references [32]. The reference co-citation network from 2012–2022 is shown in Figure 6, containing 622 effective nodes and 2264 links. Every node in Figure 6 showed an article cited by other literature, while the links represented their co-cited relationship. These nodes were also displayed in terms of tree rings to show the citation frequency of an article. The thickness of citation tree rings indicates the citation frequency of these articles in the corresponding time slice. The thickest nodes with tree rings indicate the most important references and lay out the starting point of related research. The article with the thickest citation rings, published in *Remote Sensing of Environment*, introduced Google’s powerful computing capability and bravely solved many social problems such as diseases, food security, and water resources management [33]. The highest citation frequency of this paper also indicated that the application of remote sensing technology in food safety had been one of the hot issues in recent years.

#### 4.2.3. Citation Burst Analysis

Citation burst analysis [34] was used to search the representative articles with high citation growth rates. A citation burst is an abrupt increase in citations for an article. For example, if the article’s citations suddenly increased in recent years, this reference had a strong citation burst [35]. As shown in Figure 7, the articles with strong influences signified the new pivotal turning points in the subfield research and indicated profound prospects and future trends. The red lines represented the time range of a reference citation burst, and the strength index meant the citation growth rate.

Among the 15 references, the burst strength of LeCun et al. (2015) [3] was 12.05, the strongest during the period from 2018 to 2020. This paper provided a clear introduction and profound interpretation of deep learning, which laid a strong theoretical foundation for subsequent research on deep learning and made a great contribution to research on AI in food safety. In addition, Schmidhuber (2015) summarized DL in the supervised and unsupervised neural networks, feedforward, and recurrent neural network [36]. 

In the subfield of food science technology, hyperspectral and multispectral imaging were proven to be effective non-destructive detection techniques for assessing food quality objectively and accurately and may be of interest in computer vision applications for the precise prediction of bacterial loads [37,38,39,40,41,42].

Furthermore, AI applications can be classified into two types: food safety evaluation and authenticity claims [43]. In the subfield of nutrition dietetics, Zeevi et al. (2015) designed an algorithm that combined personal metrics with behaviour habits to customize dietary intake by predicting glycemic responses [44]. In the subfield of remote sensing, Mosleh et al. (2015) reviewed the fusion of optical images, microwave technologies, and AI methods for mapping areas and forecasting yield [45]. In the subfield of environmental sciences, Xiong et al. (2017) proposed a comprehensive approach using pixel-based classification and object-based segmentation for mapping the geographical extent of croplands, which was of great importance for field management [46]. In the subfield of imaging science information systems, Simonyan and Zisserman (2015) [17] proposed two advanced CNN models to improve accuracy, while He et al. (2016) [16] presented a novel algorithm to simplify the training of CNNs. Recently, machine learning has been increasingly applied to explore the rapidly increasing foodborne pathogen genome resources and their metadata. Tree boosting is a highly effective machine-learning method and has been widely used by researchers [47].

#### 4.2.4. Journal Co-Citation Analysis

Journal co-citation analysis revealed the structure and distribution of knowledge by displaying the network of the co-cited journals. As shown in Figure 8, there were 30 journals containing 159 effective nodes and 845 links. Every node represents a journal in the datasets. These nodes were plotted in terms of tree rings to show the citation frequency of a journal. The thickness of the citation tree rings indicates the citation frequency of these journals in the corresponding time slice. The links between these nodes display the co-citation relationships of different journals. The most frequently cited journals among the selected datasets are *PLoS ONE*, *Computers and Electronics in Agriculture*, *Nature*, *Remote Sensing*, and *Scientific Reports*. From 2019 to 2022, the growth rate of citations of these five journals was significantly higher than before, as shown in Figure 9.

### 4.3. Co-authorship Analysis

The institute and country co-authorship network revealed the cooperative relationships among different countries and academic units [48]. As shown in Figure 10, there were 391 effective nodes and 582 links, including the top 50 institutes. These nodes were plotted in terms of tree rings—the thicker the tree rings, the more active the institutes. As shown in Figure 11, the top five institutions in terms of the number of publications were Chinese Acad Sci, Univ Chinese Acad Sci, China Agr Univ, Zhejiang Univ, Chinese Acad Agr Sci, etc. The results of co-analysis by institutions and countries indicated that China had paid much attention to AI applications in food safety.

## 5. Keywords and Hot Spots

### 5.1. Keyword Co-Occurrence Analysis

In order to explore the topological features and historical trajectory of a single theme or area, we can analyse the evolution of co-keyword networks as well as keyword co-occurrence networks [49]. We used keyword time-zone visualization to track the themes of AI research and depicted the hot spots in each time slice, as shown in Figure 12. We selected the keyword as the node type in CiteSpace and set the TopN to 50 and the time slice to 1. The year under keywords from 2012 to 2022 indicated when the hot spots first appeared. In each column, the nodes were arranged from bottom to top in descending order of frequency. Based on the keyword time zone map, we divided the application of AI in the food field into three stages.

Stage 1 (2012–2014): The initial stage of the AI algorithm model constructed in food safety. In this stage, hyperspectral imaging technology combined with AI models was used for food quality assessment and non-destructive testing of food products’ internal and external characteristics. The artificial neural network (ANN) algorithm was gradually applied to deal with image data collected by near-infrared spectroscopy and other instruments [50]. In the first stage, the high-frequency words, including classification, model, identification, etc., indicated the relevant algorithms were used to build models to predict crop yields or classify products.

For instance, compared with the widely used partial least squares regression (PLS), a combined strategy of back propagation artificial neural network (BP-ANN) and genetic algorithm (GA) was proven to be much more efficient [51]. In this case, the root-mean-square error (RMSE) was compared with the results of near-infrared spectroscopy (NIR) after calculating the spectra data. If RMSE was unacceptable, it would return to the input layer until the RMSE was less than the pre-set value. The RMSE of the prediction (RMSEP) between the measured and predicted values is estimated as follows:(1)$$\mathrm{RMSEP}=\sqrt{\frac{\Sigma_{i}{\left({\hat{y}}_{i}-y_{i}\right)}^{2}}{n}}$$

Stage 2 (2015–2017): The flourishing period of machine learning applied in the food industry. ML, accompanied by big data and remote sensing technologies, was widely used in food nutrition, yield prediction, and the agricultural environment. In addition, DL-based visual recognition algorithms and edge computing-based service computing paradigms were developed for dietary assessment [52]. The high-frequency words included machine learning, system, big data, remote sensing, etc. In general, the large-scale research on and applications of ML in this period provide ideas and inspiration for the next stage.

For instance, two ML models, model-based recursive partitioning (MOB) and Bayesian neural networks (BNN), were applied in forecasting crop yields for the Canadian Prairies [53]. Especially if there were more years with dramatically variable yield data, the MOB and BNN functions would likely identify the slight nonlinear relationship and thus outperform the linear techniques such as multiple linear regression (MLR). In this BNN model, the yield was expressed as a nonlinear function:(2)$$\hat{y}=\underset{j}{\sum }{\tilde{w}}_{j}\tanh\left(\Sigma_{i}w_{ji}x_{i}+b_{j}\right)+\tilde{b}$$
where $\hat{y}$ was the model output, tanh denoted the hyperbolic tangent function, and the parameters,$\;{\tilde{w}}_{j}$,$\tilde{\;b}$,$w_{ji}$, and $b_{j}$, were determined by fitting the nonlinear function to the data.

Stage 3 (2018–2022): The boom period of deep learning applied in precision food safety. As a branch of ML, deep learning disrupted many food safety domains, from algorithms to architectures, from precision agriculture to precision nutrition [54,55]. The high-frequency keywords at this stage were deep learning, artificial intelligence, convolutional neural network, etc. Moreover, advances in DL have made great contributions to integrated precision food safety across industry, research, and healthcare.

A case in point is a method to detect coffee adulteration. Based on the previously trained ResNet (He et al., 2016) [16], the convolutional algorithms with transfer learning were designed to reduce the required images and final mathematical model costs. The models can distinguish different types of coffee with errors below 1.0% and detect adulterations below 1.4%, thus benefiting producers, distributors, and consumers [56]. Formally, the building block of ReNet was defined as:(3)$$y=ℱ\left(x,\left\{w_{i}\right\}\right)+x$$

Here x and y were the input and output vectors of the layers considered. The function $ℱ\left(x,\left\{w_{i}\right\}\right)$ represented the residual mapping to be learned.

### 5.2. Research Hotspots

Through cluster analysis of the keyword, we summarized the research hotspots of AI technologies in food safety. With CiteSpace, we selected the keyword as the node type and set the TopN to 50 to analyse the top 50% with the highest frequency each year. In addition, the year of each slice was set to 1, and the threshold value to (2, 2, 20) (4, 3, 20) (3, 3, 20). Seven clusters were obtained: remote sensing, food quality, personalized nutrition, big data, food safety, deep learning, and artificial neural networks.

As shown in Figure 13, N = 494, E = 3146, density = 0.0258, modularity Q = 0.4477 (>0.3), and silhouette S = 0.7082 (>0.7), indicating significant clustering structure and the fit goodness of the graph [57]. Popular topics, keywords, authors, and journals of 143 highly cited and high-quality articles are listed in Table 1.

#### 5.2.1. Cluster #0—Remote Sensing

The name of Cluster #0 was remote sensing. Remote sensing data were usually integrated with socio-economic factors, soil data, and climate data properties based on the Google Earth Engine platform to build ML or DL models for predicting yield. Compared to traditional ground-based field surveys, empirical statistical models, and crop growth models, ML and DL methods improved yield prediction accuracy at low cost and thus were significant to food security and policymaking, particularly in the continually changing environments of population and climate [58,61,63]. In addition, remote sensing data can also be applied to crop monitoring, mapping, and classification by random forest (RF) modelling [59,60,62,64].

#### 5.2.2. Cluster #1—Food Quality

The name of Cluster #1 was food quality. Non-destructive testing technology has been crucial in monitoring food quality, rapid identification, and classification. ML methods, such as support vector machine (SVM), linear discriminant analysis (LDA), back propagation artificial neural network (BP-ANN), k-nearest neighbours, J48 decision tree, and random forest (RF), can be used to evaluate the degree of freshness of meat and the grade product quality of cereal crops precisely, at the microscopic level, and have thus played a vital role in food quality control [73]. Especially in the case of walnut processing, ML techniques such as extreme learning machine (ELM) and SVM were applied to obtain the most optimal performance in foreign object identification and food quality evaluation [84].

#### 5.2.3. Cluster #2—Personalized Nutrition

The name of Cluster #2 was personalized nutrition. Nutritional diets and healthy behaviour are critical factors in preventing and controlling non-communicable diseases. Machine learning, mobile technology, and the Internet of Things (IoT) have been applied in the personalized nutrition domain to develop robust and impactful data-driven interventions [44,113,114]. A web-based expert system for nutrition diagnosis was proved to be more accurate than a human dietitian [118]. In addition, based on the rough set theory, Lei et al. (2018) proposed an improved algorithm to select the corresponding core ingredients, with the recommended food exerting positive effects on diseases [117]. Despite advances, those applications were still in their infancy, and much research was needed before they were widely applied in clinical and public health settings [112].

#### 5.2.4. Cluster #3—Big Data

The name of Cluster #3 was big data. With the rapid development of IoT, data-driven food safety governance was the subject of much scholarly research. Zhang et al. (2013) proposed an algorithm to track pollution sources and trace back potentially infected food in the markets [125]. The impact of environmental factors, such as temperature, pesticides, and rainfall, was estimated at the farm level on agricultural crop yield [130,131]. Unmanned aerial systems-based high throughput phenotyping system has become a precise and reliable platform at field scales and has even sped up breeding cycles in many crops [127]. In addition, big data gathered from social media was analysed for consumer behaviour and used for food product development [126]. Augmented/mixed reality technologies may integrate with other emerging techniques to become one interesting future direction in the food sector [134].

#### 5.2.5. Cluster #4—Food Safety

The name of Cluster #4 was food safety. Salmonella is an important factor causing foodborne illness, while the prediction models based on ML or DL play an increasingly significant role in the assessment of food safety. To eliminate Salmonella in ground chicken, Oscar (2017) developed a multiple-layer feedforward neural network model [140]. An automated food processing line guided by AI was conceptually designed to improve the microbial detection and quality evaluation of liquid foods [141]. Virtualization and mathematical modelling represented a new and sophisticated strategic tool for designing and innovating the food processing system [142].

#### 5.2.6. Cluster #5—Deep Learning

The name of Cluster #5 was deep learning (DL). DL has been widely applied in food safety, from farm practices to dietary intervention. DL architectures provide critical tools for crop disease prediction and automated agricultural farm monitoring [144,149]. Xiao et al. (2022) introduced the application of DL in food detection systems from hardware to software [151]. Liu et al. (2021) introduced a CNN model to feature extractors for complex food, such as cereals, meat and aquatic products, fruits, and vegetables [145]. Zhu et al. (2021) reviewed the application of traditional ML and DL methods, including machine vision techniques, in food processing [152]. Tay et al. (2020) discussed current dietary assessment methods, including DL applications in food volume estimation [170]. To evaluate nutrient content quickly and accurately, Shao et al. (2022) introduced a non-destructive detection method based on Swin Transformer as the backbone network for image feature extraction [153].

#### 5.2.7. Cluster #6—Artificial Neural Network

The name of Cluster #6 was the artificial neural network (ANN). Without complete information and even any prior knowledge, ANN models can still identify nonlinear relationships and predict dependent response, and thus were applied in food image analysis, quality detection, food safety risk prediction, crop distribution, and yield prediction, and various thermal and non-thermal food-processing operations [173,177,179,182,188,191,197]. Geng et al. (2017) introduced a predictive model based on AHP integrated extreme learning machine (ELM), rather than a traditional artificial neural network (ANN), to monitor the food safety system in China [193]. Pham et al. (2020) and Anandhakrishnan and Jaisakthi (2022) separately detected early diseases on plant leaves with small disease spots through ANN and deep convolutional neural network (DCNN) [176,180]. Zhao et al. (2022) proposed a hybrid convolutional network combined with hyperspectral imaging for wheat seed classification [178]. Fu et al. (2022) employed a radial basis function neural network model to detect heavy metal contents in saline–alkali land [198].

## 6. Discussion

### 6.1. Theoretical Implications

Next-generation food systems should provide high-quality, nutritious food in a more sustainable way with regard to molecular breeding, agricultural production, food processing and distribution, and food nutrition [199]. After re-screening the 143 articles obtained in the previous section, we selected a total of 28 articles and divided them into the above four application research areas to explore the roadmap for the future. Table 2 lists the AI techniques used in each subfield and the titles of related articles.

#### 6.1.1. AI Technologies in Molecular Breeding

Since traditional breeding technologies are limited, laborious, and time-consuming selection processes, AI accelerated the breeding cycle [88,128]. ML was used as a prediction model to identify different patterns in large-scale datasets based on prior knowledge [111,200]. For instance, Zhao et al. (2020) [135] chose the most optimal hyperparameters and kernel function for the SVM model to explore the genomic-based prediction performance in pigs and maize. To gain the possible significant loci, Liu et al. (2020) constructed the maize gene, SNP locus, and carotenoid components network using the conditional Gaussian Bayesian network learning method [120]. Lv et al. (2022) proposed minmax concave penalty (MCP) regularization for sparse deep neural networks (DNN-MCP), which can provide the optimal sparse structure for DNN and then greatly improve their ability to predict genomes, especially for the genomes of three quantitative traits [187]. Recently, a software package MODAS (multi-omics data association analysis), applied dimensionality reduction (DR) to accelerate association analysis of genotypic data [137]. ML and DL have played increasingly crucial roles in exploiting multi-omics data and discovering knowledge of molecular breeding [201].

#### 6.1.2. AI Technologies in Agricultural Production

Sustainable agricultural production is critical to food security, considering the growing populations and limited resources. ML methods, such as SVM, RF, and KNN, have been used to forecast agricultural production and productivity and guide precision fertilization [79,80,81,91,93,94,189]. Deep learning, especially CNN-based image processing, was combined with gene technology, remote sensing, cloud computing, and IoT to improve the efficiency and effectiveness of the food supply chain in the whole process [139,202]. Several examples indicated applications of CNN models to monitor and predict leaf diseases [146,148,157,176].

Recent successful cases include the application of XGBoost in the fields of crop yield prediction and oyster culture environment monitoring [95,203]. On the other hand, RNN, BNN, and some ML models, such as SVM, SVR, RF, and ANFIS, have been employed for yield predictions [65,76,204]. In addition, more and more model predictive control methods involving agricultural infrastructure, field management, product processing, and greenhouses promote the transformation from conventional agriculture to precision agriculture [205].

#### 6.1.3. AI Technologies in Food Processing and Distribution

Food safety accidents may be caused by cross-contamination in food processing and distribution facilities, and there is much room for future AI applications in food traceability systems [65,86,115,132]. A computer vision system based on CNN or ML models, such as SVM, KNN, J48, and RF, has been seen as a potential technique for automatic food classification, adulterant quantification, and feature extraction [72,84,103,104,145,158,160,161]. ML algorithms have also been used to improve the effectiveness of the drying system for orange slices [82]. Swarm intelligence (SI), a subfield of AI, was employed to provide an efficient approach to fresh food distribution [133]. The assessment and monitoring model developed by SVM can predict the risks incurred during transportation and thus improve food safety [87]. A three-objective distribution planner with an expert system outperformed the traditional cost optimization model and was applied to settle the distribution of fresh foods [184]. Another BN-based dynamic unsupervised anomaly detection model suggested that severe changes in related domains of the food supply chain may lead to food safety problems [102]. It can be predicted that the fusion of image processing with ML and DL from shallow to deep will create many opportunities for food processing and distribution [206].

#### 6.1.4. AI Technologies in Food Nutrition

AI technologies were also widely applied in food nutrition since dietary problems can lead to other chronic diseases and thus increase the risk of heart attacks [112]. With missing data, ML algorithms generally outperformed the statistical methods for predicting diet quality [89]. Dietary monitoring systems based on ML methods even could automatically assess dietary intake [44,113]. Shao et al. (2022) introduced a non-destructive detection method, combining Swin Transformer with a feature fusion module, which was applied to evaluate the nutrient content of food [153]. Chen et al. (2021) implemented a proprietary deep-learning model that provided a nutrition assessment of restaurant food [154]. Sundaravadivel et al. (2018) proposed a nutrition monitoring system using a Bayesian network for nutritional balance [116]. Lei et al. (2018) developed an algorithm based on rough sets to select nutritional ingredients to aid recovery from some diseases [117]. Sadhu et al. (2020) applied DE and SA algorithms, combined with the ANN-based processing model, to explore the nonlinear correlation between cooking parameters and nutritional values and found that frying time significantly impacts food nutrition [196]. Undoubtedly, ML will develop and expand new methods to reveal the relationship between food composition and nutrition [90]. Furthermore, the results obtained by a DCNN-based model for nutrient estimation and dietary assessment shed light on potential future improvements [164,165].

### 6.2. Practical Implications

This study provides vital hints for researchers and practitioners in related fields. First, our review elaborated on the development trend and current research hotspots on AI technologies in food safety and predicted the future research direction, which is helpful for researchers to clarify the state of current research and grasp the direction and focus of future research. Practitioners in related industries can gain useful information from our research, have a deep insight into the development potential of AI technology, and grasp the future development direction of the industry. Second, this review may be helpful for researchers and practitioners to comprehensively understand the application status of AI technologies in the food sector. For example, which links in the food industry tried to change the status quo of food safety? What are AI technologies being widely used to promote the development of food safety? How do these technologies apply to the food safety field? Third, this paper has elaborated on the countries, institutions, and journals that have contributed much to the research on AI technologies in food safety. The research results should be helpful to researchers and practitioners in considering where to get more cooperation opportunities and where to seek more valuable information and help.

### 6.3. Limitations

Although the database used in our bibliometrics research was powerful and widely used by many researchers, the possibility of missing data from the WoS cannot be ruled out. Moreover, all the analysis work in this paper was based on the literature samples we retrieved rather than all published articles in the field of AI. Therefore, some relevant literature samples may not be included due to the scope of WoS. In addition, with the breakthrough in AI technologies, more and more fields have begun to emphasize AI applications. Some research on AI that indirectly impacts the field of food safety may not appear in the article title, abstract, and keywords. These articles may not appear in the literature samples we have searched. Therefore, to fully interpret the future applications of AI, we suggested that subsequent researchers expand the search scope and make the literature samples for analysis more comprehensive, which is helpful for gaining some insights from the literature distributed in other fields. Furthermore, we suggested that different databases (such as Google Scholar) be used to search for literature samples in future research. A comprehensive analysis should be made of literature samples searched from multiple databases.

## 7. Conclusions

In this paper, a bibliometric review was conducted to promote the development of research on AI in food safety. Specifically, the visualization tool CiteSpace was used in this review to perform several key bibliometric analyses on literature samples retrieved from the WoS database and to explore the development status and evolution trend of AI in food safety in a visual way. 

In addition, we listed the leading AI technologies in four promising research directions and summarized the existing literature. After that, we gave a brief overview of these articles. Classification algorithms based on ML and DL greatly improve genomic prediction ability and thus have been widely used in molecular breeding [108,135,137,187]. Due to the advantages of image recognition, ML and DL have been applied to monitor diseases and predict the yield in agriculture production [65,66,67,76,80,93,146,148,150,174]. Computer vision systems based on ML or DL represented great potentiality in food processing and distribution [72,84,86,102,103,115,158,160,184]. In addition, non-destructive detection methods based on ML or DL shed light on nutrient estimation and dietary assessment [116,153,154,164,165]. 

In conclusion, our study clarified the future development direction of research on AI in food safety by systematically and comprehensively understanding the state of current research and its trends.

## Figures and Tables

**Figure 1 foods-12-01242-f001:**
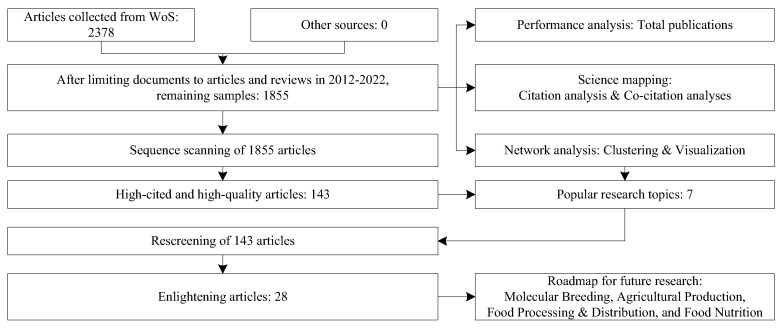
Framework of the article retrieval process and the overall research process.

**Figure 2 foods-12-01242-f002:**
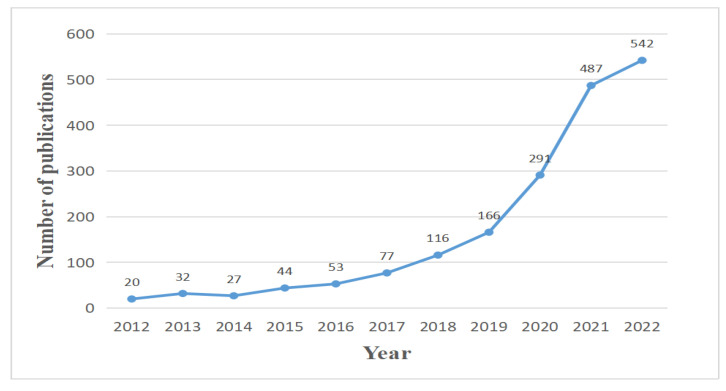
Al-related publications on food safety from 2012 to 2022.

**Figure 3 foods-12-01242-f003:**
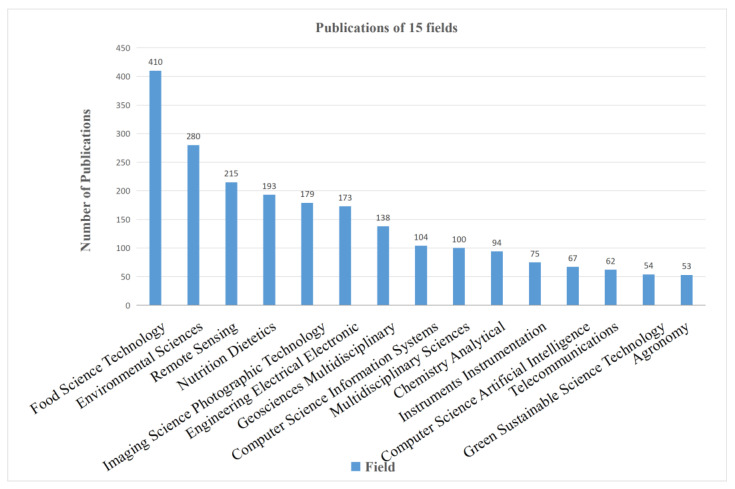
AI-related publications in different discipline categories.

**Figure 4 foods-12-01242-f004:**
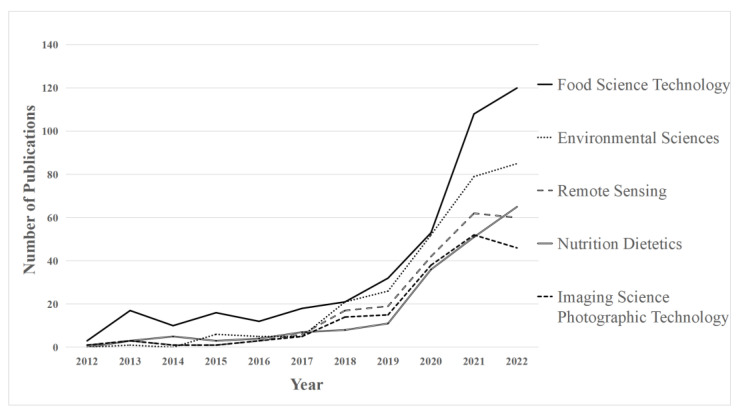
Annual publications within the five most popular subfields.

**Figure 5 foods-12-01242-f005:**
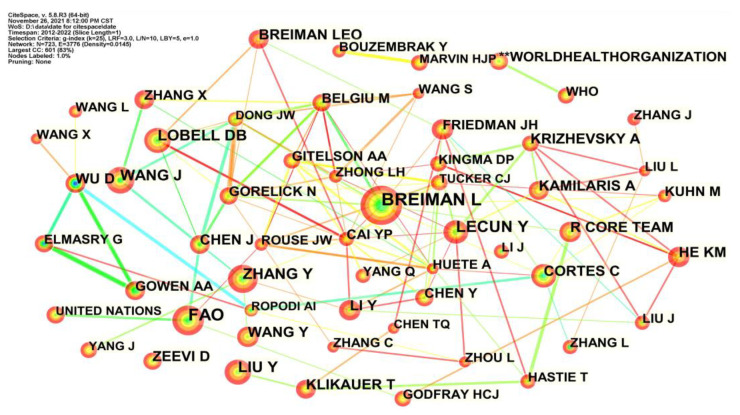
The author co-citation network of AI-related publications.

**Figure 6 foods-12-01242-f006:**
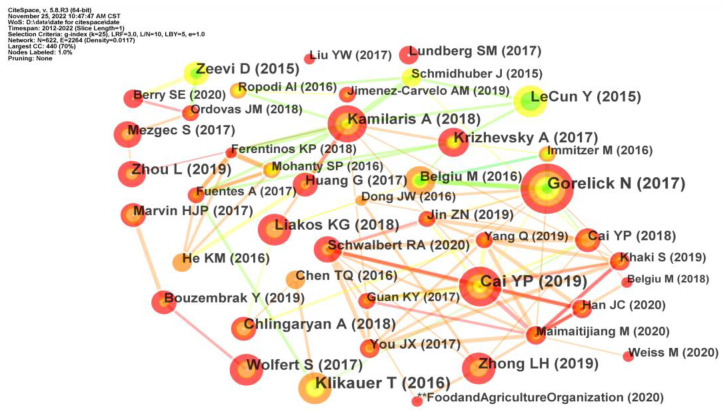
The co-citation network of AI-related publications.

**Figure 7 foods-12-01242-f007:**
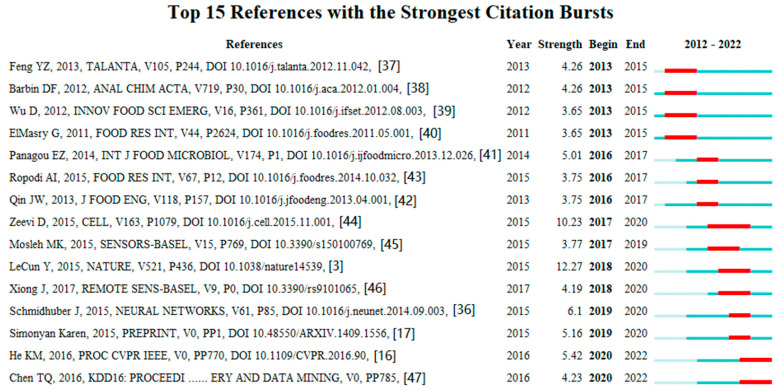
Starting point literature in related subfields.

**Figure 8 foods-12-01242-f008:**
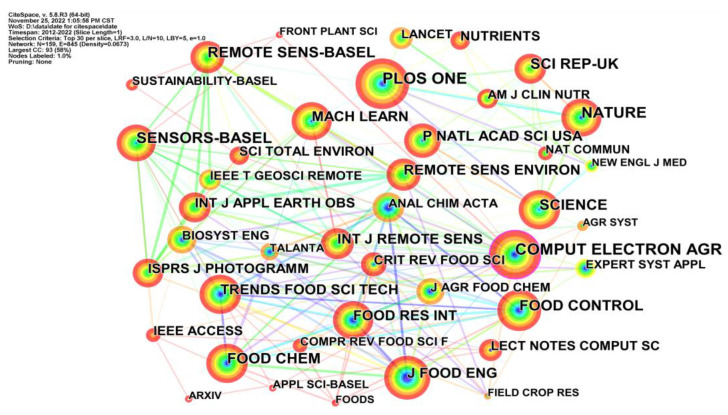
The journal co-citation network of AI related publications.

**Figure 9 foods-12-01242-f009:**
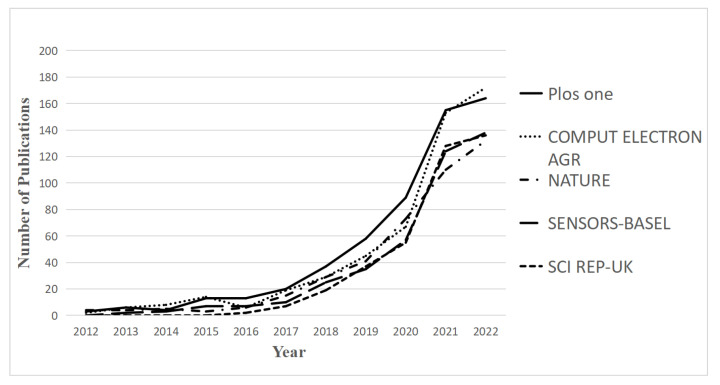
The number of co-citation articles of the top 5 journals in 2012~2022.

**Figure 10 foods-12-01242-f010:**
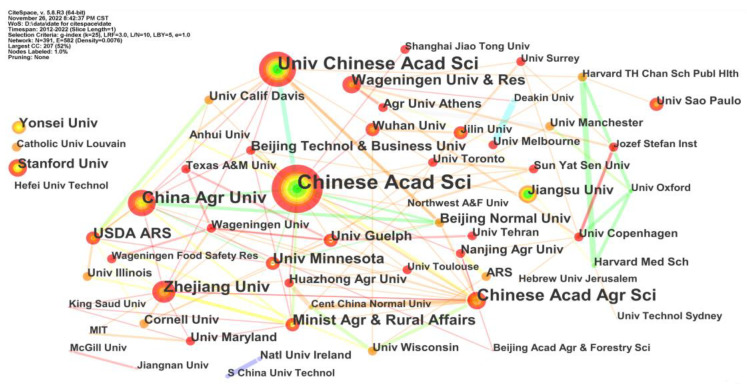
The institute co-authorship network of AI-related publications.

**Figure 11 foods-12-01242-f011:**
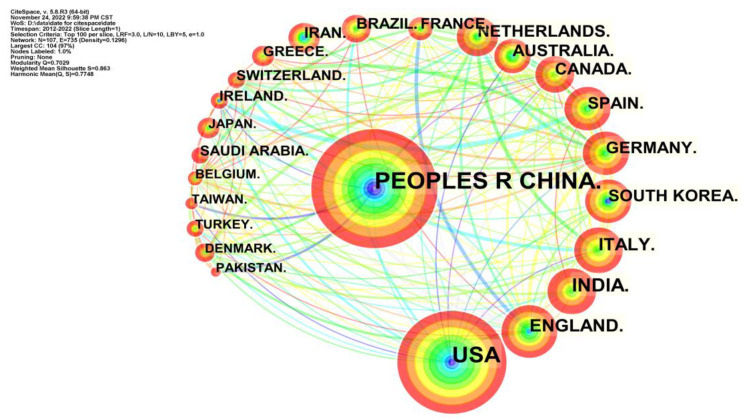
The country co-authorship network of AI-related publications.

**Figure 12 foods-12-01242-f012:**
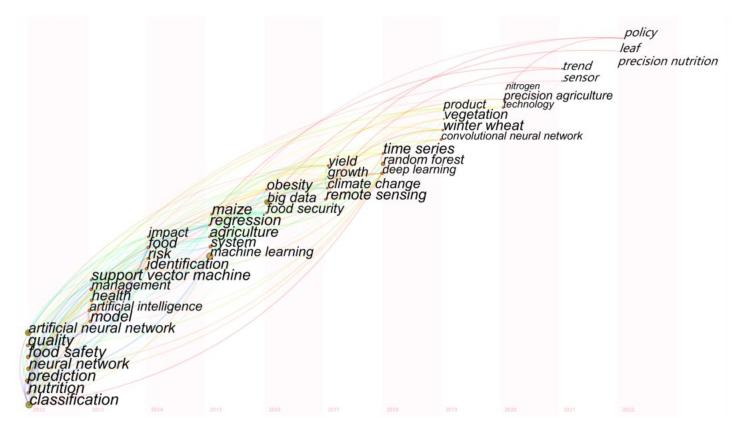
The keyword time zones of AI-related publications in 2012–2022.

**Figure 13 foods-12-01242-f013:**
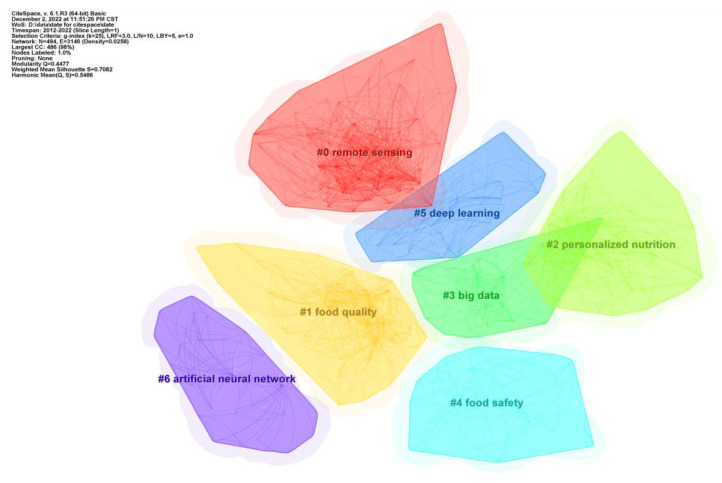
Keyword cluster analysis.

**Table 1 foods-12-01242-t001:** Topics, keywords, authors, and journals.

Topic	Keyword	Author(s)	Journal
Remote sensing	Food security	Maimaitijiang et al., 2019 [58]	1. *Remote Sensing of Environment*
	Random forest	Hao et al., 2015 [59]	2. *Remote Sensing*
		Vreugdenhil et al., 2018 [60]	3. *Remote Sensing*
	Impact	Han et al., 2020 [61]	4. *Remote Sensing*
	Climate change	Teluguntla et al., 2018 [62]	5. *Isprs Journal of Photogrammetry and Remote sensing*
		Cao et al., 2020 [63]	6. *Remote Sensing*
	Time series Vegetation Index	Duke et al., 2022 [64]	7. *International Journal of Remote Sensing*
		Ma et al., 2021 [65]	8. *Remote Sensing of Environment*
	Remote sensing	Hu et al., 2021 [66]	9. *Remote Sensing of Environment*
		Tao et al., 2019 [67]	10. *Remote Sensing*
Food quality	Machine learning	Saha and Manickavasagan, 2021 [68]	11. *Current Research in Food science*
		Jiménez-Carvelo et al., 2019 [69]	12. *Food Research International*
	Classification	Ropodi et al., 2016 [43]	13. *Trends in Food Science & Technology*
		Pu et al., 2014 [70]	14. *Meat Science*
	Prediction	Bhargava and Bansal, 2021 [71]	15. *Journal of King Saud University-Computer and Information Sciences*
		Lopes et al., 2019 [72]	16. *Sensors*
		Barbon et al., 2018 [73]	17. *Journal of Spectroscopy*
	Identification	Kim et al., 2013 [74]	18. *Food Chemistry*
		Lin et al., 2022 [75]	19. *Critical Reviews in Food Science and Nutrition*
		Crusiol et al., 2022 [76]	20. *Precision Agriculture*
	Food quality	Talukdar et al., 2022 [77]	21. *Agricultural Systems*
		Liao et al., 2021 [78]	22. *Theoretical and Applied Climatology*
	Quality	Rezapour et al., 2021 [79]	23. *Sustainability*
		Guo et al., 2021 [80]	24. *Ecological Indicators*
		Löw et al., 2018 [81]	25. *GIScience & Remote Sensing*
	Support Vector Machine	Çetin, 2022 [82]	26. *Journal of Food Processing and Preservation*
		Meenu et al., 2021 [83]	27. *Trends in Food Science & Technology*
		Magnus et al., 2021 [84]	28. *Food Control*
		Kollia et al., 2021 [85]	29. *Electronics*
		Lee et al., 2021 [86]	30. *Sensors*
		Zhang et al., 2020 [87]	31. *Sustainability*
		O’Hagan et al., 2012 [88]	32. *PLoS ONE*
		Reščič et al., 2021 [89]	33. *Nutrients*
		Barabási et al., 2020 [90]	34. *Nature Food*
		Chungcharoen et al., 2022 [91]	35. *Computers and Electronics in Agriculture*
		Habib et al. 2022 [92]	36. *Journal of King Saud University-computer and Information Science*
		Qiu et al., 2021 [93]	37. *Computers and Electronics in Agriculture*
		Kim et al., 2022 [94]	38. *Food Control*
		Ndraha et al., 2021 [95]	39. *Food Control*
		Parent et al., 2021 [96]	40. *PLoS ONE*
		Hengl et al., 2021 [97]	41. *Scientific Reports*
		Mangmee et al., 2020 [98]	42. *Food Control*
		Bouzembrak et al., 2019 [99]	43. *Food Control*
		Atas et al., 2012 [100]	44. *Computers and Electronics in Agriculture*
		Saetta et al., 2023 [101]	45. *Food Control*
		Liu et al., 2022 [102]	46. *Food Control*
		Shen et al., 2022 [103]	47. *Food Control*
		Cardoso and Poppi, 2021 [104]	48. *Food Control*
		Alfian et al., 2020 [105]	49. *Food Control*
		Davies et al., 2021 [106]	50. *Nutrients*
		Westhues et al., 2021 [107]	51. *Frontiers in Plant Science*
		Yan et al., 2021 [108]	52. *Genome Biology*
		Shete et al., 2020 [109]	53. *Plant Phenomics*
		Ni et al., 2016 [110]	54. *Frontiers in Plant Science*
		Ma et al., 2014 [111]	55. *Plant Cell*
Personalized nutrition	Nutrition	Zeevi et al., 2015 [44]	56. *Cell*
		Wang and Hu, 2018 [112]	57. *Lancet Diabetes & Endocrinology*
	Risk	Triantafyllidis, and Tsanas, 2019 [113]	58. *Journal of medical Internet research*
		Zmora and Elinav, 2021 [114]	59. *Nutrients*
	Health	Alfian et al., 2017 [115]	60. *Journal of Food Engineering*
		Sundaravadivel et al., 2018 [116]	61. *IEEE Transactions on Consumer Electronics*
		Lei et al., 2018 [117]	62. *IEEE Access*
	Data mining	Chen et al., 2012 [118]	63. *Expert Systems with Applications*
		Guo et al., 2019 [119]	64. *Molecular Plant*
		Liu et al., 2020 [120]	65. *IEEE Access*
	Disease	Wang and Yue, 2017 [121]	66. *Food Control*
		Kirk et al., 2022 [122]	67. *Advances in Nutrition*
		Gunasekara et al., 2018 [123]	68. *Nucleic Acids Research*
Bigdata	System	Frelat et al., 2016 [124]	69. *Proceedings of The National Academy of Sciences of The United States of America*
		Zhang et al., 2013 [125]	70. *International Journal of Distributed Sensor Networks*
	Model	Misra et al., 2020 [126]	71. *IEEE Internet of Things Journal*
		Jung et al.,2021 [127]	72. *Current Opinion in Biotechnology*
		Rai, 2022 [128]	73. *Molecular Biology Reports*
		Yu et al., 2013 [129]	74. *BMC Genomics*
	Big data	Al-Adhaileh and Aldhyani, 2022 [130]	75. *PEERJ Computer Science*
		McLennon et al., 2021 [131]	76. *Agronomy Journal*
		Kumar et al., 2021 [8]	77. *Journal of Food Quality*
	Artificial intelligence	Qian et al., 2020 [132]	78. *Critical Reviews in Food Science and Nutrition*
		Katiyar et al., 2022 [133]	79. *Journal of Food Quality*
		Chai et al.,2022 [134]	80. *Trends in Food Science & Technology*
	Management	Zhao et al., 2020 [135]	81. *Frontiers in Genetics*
		Khan et al., 2020 [136]	82. *Sensors*
		Liu et al., 2022 [137]	83. *Science Bulletin*
		Morgenstern et al., 2021 [138]	84. *Advances in Nutrition*
Food Safety	Food safety	Kittichotsatsawat et al., 2021 [139]	85. *Sustainability*
	Growth	Oscar, 2017 [140]	86. *International Journal of Food Science and Technology*
	Network	Kyaw et al., 2022 [141]	87. *Critical Reviews in Food Science and Nutrition*
	Temperature	Erdogdu et al., 2017 [142]	88. *Food Engineering Reviews*
		Nogales et al., 2022 [143]	89. *Food Control*
Deep learning	Deep learning	Zhou et al., 2019 [13]	90. *Comprehensive reviews in food science and food safety*
		Zhang et al., 2019 [144]	91. *Remote Sensing*
		Liu et al., 2021 [145]	92. *Trends in Food Science & Technology*
		Kaur et al., 2022 [146]	93. *Sensors*
		Wolanin et al., 2020 [147]	94. *Environmental research letters*
		Wongchai et al., 2022 [148]	95. *Ecological Modelling*
		Hu et al., 2020 [149]	96. *IEEE Access*
	Feature extraction	Zambrano et al., 2018 [150]	97. *Remote Sensing of Environment*
		Xiao et al., 2022 [151]	98. *Frontiers in Nutrition*
		Zhu et al., 2021 [152]	99. *Current Research in Food Science*
		Shao et al., 2022 [153]	100. *Foods*
		Chen et al., 2021 [154]	101. *Nutrients*
		Veeramani et al., 2018 [155]	102. *BMC Bioinformatics*
		Zhai et al., 2022 [156]	103. *PLoS ONE*
	Image	Dey et al., 2022 [157]	104. *Computers and Electronics in Agriculture*
		Rong et al., 2019 [158]	105. *Computers and Electronics in Agriculture*
		Too et al., 2019 [159]	106. *Computers and Electronics in Agriculture*
		Chakravartula et al., 2022 [160]	107. *Food Control*
		Estrada-Pérez et al., 2021 [161]	108. *Food Control*
		Vo et al., 2020 [162]	109. *Food Control*
		Izquierdo et al., 2020 [163]	110. *Food Control*
	Convolutional neural network	Hafiz et al., 2022 [164]	111. *Journal of King Saud University-Computer and Information Sciences*
		Ma et al., 2021 [165]	112. *Food Research International*
		Ahn et al., 2019 [166]	113. *Sensors*
		Yang et al., 2021 [167]	114. *Plant Methods*
		Zingaretti et al., 2020 [168]	115. *Frontiers in Plant Science*
		Ma et al., 2018 [169]	116. *Planta*
		Tay et al., 2020 [170]	117. *Nutrients*
Artificial neural network	Neural network	Huang et al., 2014 [171]	118. *Food Chemistry*
		Delloye et al., 2018 [172]	119. *Remote Sensing of Environment*
		Das et al., 2018 [173]	120. *International Journal of Biometeorology*
		Geng et al., 2017 [174]	121. *Food Control*
		Al-Mahasneh et al., 2016 [175]	122. *Food Engineering Reviews*
		Anandhakrishnan and Jaisakthi 2022 [176]	123. *Sustainable Chemistry and Pharmacy*
		Chamundeeswari et al., 2022 [177]	124. *Microprocessors and Microsystems*
	Artificial neural network	Zhao et al., 2022 [178]	125. *Infrared Physics & Technology*
		Sujarwo et al., 2022 [179]	126. *Sustainability*
		Pham et al., 2020 [180]	127. *IEEE Access*
		Tao et al., 2019 [181]	128. *Journal of Integrative Agriculture*
		Raj and Dash, 2022 [182]	129. *Critical Reviews in Food Science and Nutrition*
		Kondakci and Zhou, 2017 [183]	130. *Food and Bioprocess Technology*
		Bortolini et al., 2016 [184]	131. *Journal of Food Engineering*
		Okut et al., 2013 [185]	132. *Genetics Selection Evolution*
		González-Camacho et al., 2012 [186]	133. *Theoretical and Applied Genetics*
	Performance	Lv et al., 2022 [187]	134. *Genomics*
		Li et al., 2022 [188]	135. *Food Control*
		Shi et al., 2021 [189]	136. *Computers and Electronics in Agriculture*
		Kuzuoka et al., 2020 [190]	137. *Food Control*
		Tao et al., 2020 [191]	138. *Sensors*
		Tian et al., 2020 [192]	139. *Computers and Electronics in Agriculture*
		Geng, 2019 [193]	140. *Food Control*
		Wang et al., 2017 [194]	141. *Food Control*
		Silva et al., 2015 [195]	142. *Food Control*
		Sadhu et al., 2020 [196]	143. *Journal of Food Process Engineering*

**Table 2 foods-12-01242-t002:** AI technologies and applications.

Field	Sample	Functionality	Method(s)	Result(s)
Molecular Breeding	Crops (Yan et al., 2021) [108]	Genomic prediction	LightGBM	LightGBM exhibited superior performance of genomic selection prediction.
	Maize (Liu et al., 2022). [137]	Germplasm exploitation	MODAS	MODAS can accelerate association analysis of genotypic data.
	Pig and maize (Zhao, et al., 2020) [135]	Genomic prediction	SVM	The prediction model based on SVM outperformed BayesR and GBLUP in two data sets.
	Sea cucumber (Lv et al., 2022) [187]	Genomic prediction	DNN-MCP RR-GBLUP Bayes B DNN	DNN-MCP can greatly improve genomic prediction ability.
Agricul-tural Production	Tomato (Anandhakrishnan and Jaisakthi, 2022) [174]	Leaf disease recognition	DCNN	DCNN model gained an accuracy of 98.40% for the testing set.
	Farm crop (Wongchai et al., 2022) [148]	Crop disease prediction	DAL_CL RNN	Experimental results showed an accuracy of 96%.
	Rice (Qiu et al., 2021) [93]	Nitrogen Nutrition Index	AB, ANN, KNN, PLSR, RF SVM	The RF algorithms performed the best, with the R^2^ ranging from 0.88 to 0.96 and RMSE ranging from 0.03 to 0.07.
	Soybean (Crusiol et al., 2022) [76]	Monitoring of yield	PLSR, SVR	Field-based SVR models presented the highest accuracies for yield mapping.
	Grape leaves (Kaur et al., 2022) [146]	Identification of leaf diseases	Hy-CNN TL, LR	For leaf disease recognition, Hy-CNN has the highest accuracy of 98.7%.
	Corn (Ma et al., 2021) [65]	Prediction of corn yield	BNN	The BNN model can predict corn yield in normal and abnormal years with extreme weather.
	Crop (Hu et al., 2021) [66]	Crop type mapping	RF-r	The spatial consistency between the sub-pixel crop distribution map generated by temporal MODIS and the medium-high resolution reference map reached 0.75.
	Rice (Guo et al., 2021) [80]	Yields prediction	MLR, BPNN, SVM, RF	In yield predictions, SVM obtained the highest precisions.
	Crop (Tao et al., 2019) [67]	Cropping intensity mapping	BNPK	For cropping intensity index mapping, BNPK model can settle intra-class variations.
	Agricultural productivity (Zambrano et al., 2018) [150]	Prediction of agricultural productivity	OLR, DL	OLR, compared to DL, only showed a slightly smaller accuracy.
Food Processing and Distribu-tion	Antioxidant peptide (Shen et al., 2022) [103]	Feature extraction	LR, LDA, SVM, KNN	The ML-based predictor was effective in mining the multifunctional peptides.
	Walnut (Rong et al., 2019) [158]	Objectives detection	CNN	The proposed method obtained an accuracy of 95% for foreign object detection.
	Walnut (Magnus et al., 2021) [84]	Non-destructive food classification	ELM, SVM LDA, QDA PLS-DA	An ML-based algorithm, compared to classical techniques, improved the performance metric by up to 80%.
	Bacterial biofilms (Lee et al., 2021) [86]	Detection of bacterial biofilms	DT, KNN LDA, PLS-DA	KNN algorithm proved a high performance in predicting the biofilm region.
	Barley flour (Lopes et al., 2019) [72]	Barley flour classification	CVS, SVM, KNN, J48, RF	The accuracy of this method ranged from 75.00% to 100.00%.
	Milk (Liu et al., 2022) [102]	Anomaly detection	BN	Food safety problems in the supply chain could be predicted by detecting severe changes in related fields.
	Coffee (Chakravartula et al., 2022) [160]	Coffee adulterant quantification	CNN	The results confirmed the feasibility of the CNN algorithm with excellent performances (R^2^ > 0.98).
	Kimchi supply chain (Alfian et al., 2017) [115]	Food traceability system	MLP	In the case of missing sensor data, MLP proved to be the best model with high prediction accuracy.
	Fresh food (Bortolini et al., 2016) [184]	Fresh food distribution	LP	The expert system outperformed the traditional cost minimization model.
Food Nutrition	Daily diet (Shao et al., 2022) [153]	Nutritional evaluation	ST, FFM	Swin-Nutrition provided a novel non-destructive detection technology.
	Restaurant food (Chen et al., 2021) [154]	Nutrition assessment.	Calorie Mama (DL model)	The DL model obtained an accuracy of 75.1%.
	Soft drinks (Hafiz et al., 2022) [164]	Classification and dietary assessment	DCNN with transfer learning	The DCNN-based transfer learning model showed an accuracy of 98.51%.
	Infant diet (Sundaravadivel et al., 2018) [116]	Automated nutrition monitoring	Bayesian network	Smart-Log predicted 8172 foods for 1000 meals with 98.6 percent accuracy.
	Chinese dishes (Ma et al., 2021) [165]	Nutrient estimation	DCNN	The DCNN model showed the highest performance for protein estimation.

## Data Availability

Data are contained within the article.

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
