# Peer review of "Artificial Intelligence in Food Safety: A Decade Review and Bibliometric Analysis"

_foods, 2023, doi:10.3390/foods12061242_

Round 1

Reviewer 1 Report

The submitted review article explores the role of artificial intelligence in food safety and its potential to improve food yield, quality, nutrition, safety, traceability, and reduce waste. The authors conducted a bibliometric analysis of 1855 articles published between 2012 and 2022 using CiteSpace to identify historical and future trends in AI research in food safety. The review provides an impressive overview of the current state of AI research in food safety, identifies research hotspots, and highlights future trends.

The authors' approach, which uses a bibliometric analysis to evaluate the research landscape, provides a comprehensive view of the evolution of AI technologies in food safety. The analysis includes performance, science mapping, and network analysis, which allows for a detailed examination of research trends and the identification of knowledge gaps.

The review also highlights the potential of AI to improve food safety across the entire food production process, from precision agriculture to precision nutrition. The authors suggest that AI could enhance the efficiency of food production by reducing resource consumption and food waste, while also increasing the safety and traceability of food products. The authors' review offers valuable insights into the role of AI in food safety, making it a helpful resource for researchers, practitioners, and policy-makers in the field.

Overall, the review's bibliometric approach provides a unique perspective on the research landscape, highlighting trends and gaps in knowledge that can inform future research efforts. The article is well-written, with clear and concise language, making it accessible to a wide audience. The review offers valuable insights and is recommended for anyone interested in the field of food safety and the role of AI in enhancing food production. Specific comments:

Page5, Page 10, The figures included in the paper are informative and provide valuable visual representations of the authors' findings. However, it is essential to note that the font size used in some figures is relatively small and may be difficult for readers to discern. As such, it is recommended that the authors increase the font size used in their figures to ensure that all information presented is legible and accessible to readers.

Page 21, While the conclusion does provide a summary of the paper's main findings, it is quite brief and could be improved. The authors should expand on their conclusions, highlighting their main findings and the implications of their research for the field of food safety and suggesting avenues for future research.

Author Response

See the attached PDF

Reviewer 2 Report

1.      I couldn't see a dot at the end of the sentence, did the sentence end on the 140th line.

2.      As stated in the conclusion, WoS database could be given in the summary.

3.      Why was only WoS database chosen?

4.      Finally, as shown in Figure 1, 28 articles have been refined and studied to present AI applications in food safety across the entire process from precision agriculture to precision nutrition (146-148). Well, what did they find in these 28 articles, you could give it a little more detail.

5.      Again, 28 articles could have been mentioned in the abstract and conclusion.

6.      In general, it was a good work on a new and interesting subject.

Author Response

See attached PDF
